# Prognostic Factors Affecting Death in Patients with Rheumatoid Arthritis Complicated by *Pneumocystis jirovecii* Pneumonia and One-Year Clinical Course: The ANSWER Cohort Study

**DOI:** 10.3390/ijms24087399

**Published:** 2023-04-17

**Authors:** Hideyuki Shiba, Takuya Kotani, Koji Nagai, Kenichiro Hata, Wataru Yamamoto, Ayaka Yoshikawa, Yumiko Wada, Yuri Hiramatsu, Hidehiko Makino, Yo Ueda, Akira Onishi, Koichi Murata, Hideki Amuro, Yonsu Son, Ryota Hara, Toru Hirano, Kosuke Ebina, Masaki Katayama, Motomu Hashimoto, Tohru Takeuchi

**Affiliations:** 1Department of Internal Medicine (IV), Division of Rheumatology, Osaka Medical and Pharmaceutical University, Osaka 569-8686, Japan; 2Department of Health Information Management, Kurashiki Sweet Hospital, Okayama 710-0016, Japan; 3Department of Rheumatology and Clinical Immunology, Kobe University Graduate School of Medicine, Kobe 650-0017, Japan; 4Department of Advanced Medicine for Rheumatic Diseases, Graduate School of Medicine, Kyoto University, Kyoto 606-8303, Japan; 5First Department of Internal Medicine, Kansai Medical University, Osaka 573-1191, Japan; 6Rheumatology Clinic and Department of Orthopaedic Surgery, Nara Medical University, Nara 634-8521, Japan; 7Department of Rheumatology, Nishinomiya Municipal Central Hospital, Nishinomiya 663-8014, Japan; 8Department of Musculoskeletal Regenerative Medicine, Osaka University Graduate School of Medicine, Osaka 565-0871, Japan; 9Department of Rheumatology, Osaka Red Cross Hospital, Osaka 543-0027, Japan; 10Department of Clinical Immunology, Graduate School of Medicine, Osaka Metropolitan University, Osaka 545-8585, Japan

**Keywords:** *Pneumocystis jirovecii* pneumonia, rheumatoid arthritis, prognosis

## Abstract

This multicenter retrospective study aimed to clarify the prognostic factors for mortality and changes in treatment modalities and disease activities after the onset of *Pneumocystis jirovecii* pneumonia (PCP) in patients with rheumatoid arthritis (RA). Data regarding the clinical background, treatment modalities, and disease activity indicators of RA at the onset of PCP (baseline), and 6 months and 12 months after treatment were extracted. Of the 37 patients with RA-PCP (median age, 69 years; 73% female), chemical prophylaxis was administered to 8.1%. Six patients died during PCP treatment. The serum C-reactive protein (CRP) levels and the prednisolone (PDN) dose at baseline in the PCP death group were significantly higher than those in the survivor group. Multivariate analysis using a Cox regression model showed that PDN dose at baseline was a predictor of death from PCP in patients with RA. During the 12 months from baseline, the RA disease activity significantly decreased. A high dose of corticosteroids for RA may result in a poor prognosis when PCP is complicated. In the future, preventive administration techniques must be established for patients with RA who need PCP prevention.

## 1. Introduction

Rheumatoid arthritis (RA) is a chronic autoimmune disease characterized by joint inflammation and destruction [1]. Patients with RA frequently develop opportunistic infections during their clinical course, including respiratory infections, which result in a poor prognosis [2,3,4].

*Pneumocystis jirovecii* (PC) is a fungal pathogen that causes pneumonia in infancy and in compromised hosts with decreased cell-mediated immunity [5,6]. The use of corticosteroids and immunosuppressants is a risk factor for PC pneumonia (PCP), and patients who develop PCP often have a fatal course [5,7]. In patients with RA, corticosteroids and immunosuppressants, including methotrexate, are used. In addition, the use of biological disease-modifying anti-rheumatic drugs (DMARDs) and targeted synthetic DMARDs (tsDMARDs) causes patients to become immunocompromised, and this is a risk for PCP comorbidity [8,9].

Risk factors for developing PCP in patients with RA include older age, male sex, corticosteroid use, and pulmonary lesions [10]. However, poor prognostic factors for mortality, changes in treatment modalities for RA, and changes in the disease activity of RA after the onset of PCP are still unclear. Therefore, we extracted the clinical background of the patients, the disease activities of RA, and the treatment modalities for RA at the onset of PCP (baseline) and examined the poor prognostic factors of mortality. In addition, we also examined changes in the treatment modalities and in the disease activities of RA during the 12 months after the baseline. The aim of the present study is to clarify the prognostic factors of mortality in RA patients with PCP and the changes in RA disease activity and treatment modalities for RA after the onset of PCP.

## 2. Results

### 2.1. Characteristics of the Patients

The study population was selected from all patients with RA-complicated PCP (RA-PCP) in the ANSWER cohort (*n* = 37) who fulfilled the inclusion criteria. The baseline characteristics of the participants are shown in Table 1. The median age was 69.0 (63–75) years, the disease duration was 10.7 (2.77–20.4) years, and 73.0% of the patients were women. In the treatment modalities for RA, methotrexate (MTX) and prednisolone (PDN) were used in 86.5% and 67.6% of patients, respectively, and the doses were 7.0 (4.0–9.5) mg/week and 4.0 (0–9.0) mg/day, respectively. The comorbidity rates of diabetes mellitus, interstitial lung disease, lung emphysema, bronchiectasis, latent tuberculosis infection (LTBI), and lung nontuberculous mycobacterial (NTM) infection were 21.6%, 29.7%, 10.8%, 8.11%, 5.41%, and 2.70%, respectively.

As a chemical prophylaxis, atovaquone was administered to 8.1% of patients. No patient was administered sulfamethoxazole-trimethoprim (SMX/TMP) mixture.

### 2.2. Treatments for PCP

The treatment details and outcomes of patients with PCP are presented in Table 2. A TMP/SMX mixture was used in 94.6% of the patients, and the initial dose was 9 (9–12) mg/day. One g of TMP/SMX mixture contains 80 mg of TMP and 400 mg of SMX. Pentamidine inhalation, atovaquone, methylprednisolone pulse therapy, and oral prednisolone were used in 32.4%, 13.5%, 32.4%, and 94.3% of patients, respectively. The initial prednisolone dose was 64 (40–80) mg/day.

### 2.3. Comparison of the Baseline Clinical Characteristics and Treatment Regimens of Patients Who Died from PCP and Survivors

During the 12-month follow-up period, 6 patients died due to PCP, and the duration between baseline and death was 13 (6.5–24.3) days. We compared the baseline clinical characteristics and treatment regimens of the 6 patients in the PCP death group and the 31 in the survivor group (Table 3). There was no difference between the PCP death and survivor groups in terms of age, sex, disease duration of RA, disease activity indicators, or complications. Although the serum CRP levels of 15.9 (6.98–18.6) mg/dL in the PCP death group were significantly higher than those in the survivor group of 5.7 (1.85–9.56) mg/dL (*p* = 0.035), other laboratory findings did not differ between the two groups. The PDN dose of 9.0 (5.75–19.4) mg/day in the PCP death group was significantly higher than that in the survivor group of 3.0 (0–5.0) mg/day (*p* = 0.006), but other treatment modalities for RA did not differ between the two groups.

We also compared the baseline respiratory conditions, including duration of respiratory symptoms and PaO2/FiO2 ratio, between the PCP death group and the survival group. The duration of respiratory symptoms could be extracted for 14 (9–39) days in 3 PCP deaths and 7 (4–13) days in 23 survivors, and the PaO2/FiO2 ratio could be extracted for 64.6 (46.7–251.9) in all PCP deaths and 238.1 (188.8–344.2) in 21 survivors. Although there was no difference in the duration of respiratory symptoms between the two groups, the PaO2/FiO2 ratio was significantly lower in the PCP death group compared with the survival group.

### 2.4. Cox Regression Analyses of Death from PCP in Patients with RA

Serum CRP levels and PDN dose at baseline were identified as risk factors for death from PCP. To confirm these findings, we performed univariate and multivariate analyses using a Cox proportional hazards model (Table 4). Age was considered a potential confounder and adjusted as a covariate. Univariate analysis using a Cox regression model showed that the PDN dose at baseline was a predictor of death from PCP in patients with RA (*p* = 0.003). Multivariate analysis also showed that PDN dose at baseline was a predictor of death from PCP in patients with RA (*p* = 0.003).

### 2.5. Changes in Treatment Modalities for RA and Chemical Prophylaxis for PCP during the Course after PCP Treatment

Table 5 shows the comparisons of treatment modalities for RA and chemical prophylaxis modalities for PCP at baseline, and at 6 and 12 months after. The usage rate and dose of MTX were significantly lower at 6 months after and at 12 months after baseline than those at baseline. The usage rate of PDN significantly increased at 6 months after baseline compared with that at baseline, and the dose of PDN significantly decreased at 12 months after baseline compared with that at 6 months after baseline. The usage rate of bDMARDs/tsDMARDs was significantly lower at 6 months after baseline than that at baseline. The usage rate and the dose of SMX/TMP were significantly higher at 6 months after and 12 months after, compared with those at baseline.

### 2.6. Changes in Disease Activities of RA during the Course after PCP Treatment

Table 6 shows the changes in RA disease activity during the course of PCP treatment. During the 6-month period from baseline, DAS28CRP and serum CRP levels significantly decreased (*p* = 0.046 and <0.001, respectively). During the 12-month period from baseline, DAS28-CRP, SDAI, and serum CRP levels significantly decreased (*p* = 0.001, 0.012, and <0.001, respectively). There was no significant change in RA disease activity between 6 and 12 months.

## 3. Discussion

Risk factors for PCP complications in patients with RA have been reported to be older age, male sex, existing lung lesions, and the use of corticosteroids, MTX, and bDMARDs [8,11]. In Japanese patients with RA during IFX treatment, age 65 years or older, PDN use of 6 mg/day or more, and existing lung lesions were reported as risk factors for PCP complications [11]. In Japanese patients with RA during ETN treatment, age 65 years and older, existing lung lesions, and the use of MTX have also been reported as risk factors for PCP complications [12]. In the present study, the median age of the patients was 69 years, and the complication rate of lung lesions was high. In addition, the usage rates of PDN, MTX, and bDMARDs/tsDMARDs were also high at 67.6%, 86.5%, and 37.8%, respectively, which supports the results of previous reports. A previous study reported that the median serum levels of IgG were significantly lower in patients with RA-PCP during IFX treatment (1192 mg/dL) than in patients with RA without PCP (1540 mg/dL) [13]. In the present study, the median serum IgG levels also decreased to 948 mg/dL.

Pentamidine inhalation, atovaquone, and TMP/SMX have been used to prevent PCP. The need for prophylaxis against PCP has been advocated receiving immunosuppressive drugs in addition to prednisolone 20 mg/day or higher [14]. However, there are no established guidelines for specific indications, drug selection and dosage, or administration period for PCP prevention in patients with RA. Although TMP/SMX is effective in preventing PCP, its administration is recommended when the risk of developing PCP exceeds 3% from the viewpoint of side effects [8]. Moreover, the complication rate of PCP in patients with RA using bDMARDs is as low as 0.2–0.4% [15,16,17,18,19], and it does not reach the standard for TMP/SMX use. Therefore, it is necessary to further select cases for which prophylactic administration of TMP/SMX is indicated according to the risk factors of the patient’s background. Although, in the present study, any of the risk factors for PCP complications—65 years or older, PDN doses of 6 mg/day or higher, MTX use, bDMARDs/tsDMARDs use, or existing lung lesions—were present in all patients, PCP prophylaxis by atovaquone was performed in only three patients. This suggests that, in the real world, sufficient PCP prophylaxis may not be administered to patients with RA. While considering the side effects of PCP preventive drugs, it is also necessary to pay attention to the risk factors for PCP complications. Of 107 Japanese CTD patients receiving high-dose corticosteroids who were prophylactically treated with atovaquone, 5 were reported to have complicated PCP [20]. Although the frequency may be low, it is also considered necessary to pay attention to the possibility of complication with PCP even during prophylactic administration of atovaquone in patients with RA under immunosuppressive therapy.

Existing lung lesions have been reported to affect the mortality and prognosis of patients with RA-PCP [11,12]. Although there was no difference in the complication rate of existing lung lesions between the death and the survivor groups in the present study, existing lung lesions may have affected the deterioration of respiratory status due to PCP. Poor prognostic factors for mortality in patients with RA-PCP have not yet been clarified. In the present study, high-dose corticosteroids at baseline were independently associated with PCP mortality. These results suggest that the prognosis may be poor in cases of severe immunosuppression in RA-PCP. Even with diseases other than RA, it has been reported that long-term or high-dose corticosteroid use is associated with the risk of developing PCP in patients with asthma, COPD patients with lung cancer, and patients with COVID-19 infection [21,22,23,24,25,26]. It is suggested that reducing the amount and duration of corticosteroid use may reduce the complication rate of PCP and improve prognosis.

In the present study, the MTX concomitant rate/dose decreased, and the PDN concomitant rate increased 6 months after the onset of PCP. Owing to excessive immunosuppression and the risk of drug-induced pneumonia, the use of MTX might be avoided after the onset of PCP. The suppression of RA disease activity may depend on concomitant/increased doses of corticosteroids for the treatment of PCP. In the present study, ETN and ABT were the most common bDMARDs used after PCP onset. In postmarketing studies of each bDMARD in patients with RA, PCP was complicated in 0.44% of patients for IFX [15], 0.18% for ETN [16], 0.3% for ADA [17], 0.2% for TCZ [18], and 0.1% for ABT [19], respectively. Although a direct comparison is not possible due to differences in patient backgrounds in each study, ETN and ABT may be associated with a lower prevalence of PCP than other bDMARDs. Therefore, the ETN and ABT may have been selected after the PCP merger. In addition, ETN is a soluble receptor preparation and has been reported to have lower antibody and complement-dependent cellular cytotoxicity against cells expressing membrane-type tumor necrosis factor (TNF) than TNF antibody preparations, suggesting that it may reduce the risk of cell-mediated immunosuppression [27]. In addition, ABT was associated with a lower incidence of comorbid infections than other bDMARDs, and the incidence of infections during ABT use was not different from that in patients treated with csDMARDs [28]. These findings suggest that ETN and ABT are more likely to be selected as bDMARDs after PCP.

To date, details of changes in RA disease activity after PCP onset have not been reported. In the present study, comparing each RA activity index at the time of PCP onset and 12 months later, DAS28CRP and SDAI decreased significantly, but CDAI did not change. The components of DAS28CRP and SDAI include serum CRP values, which may be affected by corticosteroids for PCP treatment. Therefore, regarding the evaluation of RA disease activity after the onset of PCP, it was suggested that it might be appropriate to use the CDAI, which does not include inflammatory reactions as components. As these clinical measures alone may not be sufficient to assess RA disease activity after PCP complications, a combination of objective measures, such as joint ultrasonography, may be useful.

As this was a multicenter cohort study, there were some limitations in the clinical data extraction and interpretation of the results. First, this study was a retrospective cohort study with a small sample size, and there was a possibility of bias due to missing values regarding various data. Second, because PCP was diagnosed by each attending physician at many facilities, the details of the diagnosis and severity of PCP could not be extracted. Third, all of the research participants were Japanese, and it may not be possible to generalize the results to other races. Fourth, the survivor group was skewed, with five times as many patients as non-survivors, which could confound the results. To resolve these limitations, further studies with more races and more cases are needed in the future.

## 4. Materials and Methods

### 4.1. Patients

The Kansai Consortium for Well-being of Rheumatic Disease Patients (ANSWER) cohort is a multicenter observational registry of patients with RA in the Kansai district of Japan [29,30,31]. Data from patients at seven institutes (Kyoto University, Osaka University, Osaka Medical and Pharmaceutical University, Kansai Medical University, Kobe University, Nara Medical University, and Osaka Red Cross Hospital) were included. From August 2008 to October 2017, 6939 patients with RA were registered, and 78,256 serial disease activities were available from the database. Data regarding patients with RA-complicated PCP were retrospectively collected. RA was diagnosed based on either the 1987 RA classification criteria of the American College of Rheumatology (ACR) [32] or the 2010 ACR/European League Against Rheumatism criteria [33]. Attending rheumatologists administered the treatments.

### 4.2. Ethics Approval and Consent to Participate

The representative facility of this registry was Kyoto University, and this observational study was conducted in accordance with the Declaration of Helsinki with the approval of the ethics committees of the following seven institutes: Kyoto University (2016-03-24/approval no. R053), Osaka University (2015-11-04/approval no. 15300), Osaka Medical and Pharmaceutical University (2014-07-14/approval no. 1529), Kansai Medical University (2017-11-21/approval no. 2014625), Kobe University (2015-03-20/approval no. 1738), Nara Medical University (2018-01-23/approval no. 1692), and the Osaka Red Cross Hospital (2015-09-01/approval no. 644). At institutes other than Osaka University Hospital, written informed consent was obtained from the participants. The Osaka University Hospital Ethics Committee board waived the requirement for patient informed consent because of the anonymous nature of the data.

### 4.3. PCP Diagnosis

PCP was diagnosed by the attending physicians based on the clinical and laboratory findings. In the present study, PCP was suspected when a patient had the following clinical symptoms: fever, non-productive cough, dyspnea on exertion, relatively homogenous ground glass opacity in the bilateral lung fields on chest images, elevated serum (1-3)-β-D-glucan (βDG), and progressive hypoxia. PCP was diagnosed as positive for *Pneumocystis jirovecii* in the sputum or bronchoalveolar lavage fluid (BALF) by polymerase chain reaction (PCR) or microscopic examination [34]. A PCR method targeting the 5.8S rRNA gene and internal transcribed spacer 2 (ITS2) gene is used to detect *P. jirovecii* DNA.

### 4.4. Data Collection

Data were extracted from the patients’ medical records. In the baseline demographic data, age, sex, duration of RA, complications, and outcomes were extracted. Laboratory findings included white blood cell (WBC) and lymphocyte counts, serum albumin, creatinine, lactate dehydrogenase (LD), C-reactive protein (CRP), Krebs von den lungen 6 (KL-6), immunoglobulin G (IgG), and serum βDG levels. The respiratory conditions, including duration of respiratory symptoms and PaO2/FiO2 ratio at baseline, were also extracted. In addition, the disease activities of RA, treatment modalities for RA, prophylaxis, and treatment modalities for PCP were also extracted at baseline, at 6 and 12 months after treatment.

### 4.5. Evaluation of RA Disease Activity

Data regarding swelling joints, tender joints, the patient’s global assessment of disease activity using a visual analog scale (Pt-VAS), the physician’s global assessment of disease activity via a visual analog scale (Ph-VAS), and serum CRP levels were extracted as parameters related to RA disease activity. Data regarding the Disease Activity Score-28-CRP (DAS28-CRP), the Simplified Disease Activity Index (SDAI), the Clinical Disease Activity Index (CDAI) for RA disease activity, and the Health Assessment Questionnaire (HAQ) were also extracted [35,36,37]. The DAS28-CRP is widely used, and cut-off points of 2.3, 2.7, and 4.1 have been proposed to be indicative of remission, low disease activity, and high disease activity, respectively. With DAS28-CRP, it is possible to objectively evaluate the extent of the current medical condition and to evaluate the effect of treatment by tracking numerical values over time. The SDAI is a simpler evaluation method than the DAS28-CRP and is calculated based on the number of tender and swollen joints, the Pt-VAS, the Ph-VAS, and serum CRP levels. Moreover, the CDAI is the only composite index that does not incorporate serum CRP levels and therefore can be used to conduct a disease activity evaluation essentially anytime and anywhere. HAQ is an assessment method that evaluates comprehensive quality of life, including not only the patient’s own physical function, but also mental function, various complaints, malaise, and work conditions.

### 4.6. Statistical Analysis

Data are presented as the median (interquartile range). Fisher’s exact test was used to compare categorical variables, and the Mann–Whitney U test was used to compare median values. The Wilcoxon signed-rank test was used to compare various pre- and post-treatment parameters. We compared the demographic and background characteristics between the two groups using univariate analysis, then estimated the hazard ratios of patient outcomes in univariate and multivariate analyses using a Cox regression model. Statistical significance was set at *p* < 0.05. Data were analyzed using JMP version 14.0 (SAS Institute Inc., Cary, NC, USA).

## 5. Conclusions

In this multicenter cohort study of Japanese patients with RA, the concomitant rate of PCP preventive drugs was low in patients with PCP, and a high dose of corticosteroids may have a poor prognosis for mortality. It is necessary to select patients with RA who require PCP prevention and to establish preventive administration methods in the future.

## Figures and Tables

**Table 1 ijms-24-07399-t001:** Characteristics of the subjects at baseline.

Characteristics	All Patients (*n* = 37)
Age (years)	69 (63–75)
Female, number (%)	27 (73.0)
Disease duration (years)	10.7 (2.77–20.4)
Missing or unmeasurable	1
DAS28CRP	3.58 (2.92–4.57)
Missing or unmeasurable	18
CDAI	8.6 (2–12.2)
Missing or unmeasurable	20
SDAI	16.1 (7.44–22.5)
Missing or unmeasurable	20
HAQ score	0.81 (0–1.78)
Missing or unmeasurable	23
Laboratory findings	
WBC (/μL)	7900 (5960–10,255)
Lymphocytes (/μL)	800 (550–1476)
Serum alubmin (mg/dL)	3.2 (2.75–3.5)
Serum creatinine (mg/dL)	0.80 (0.63–1.02)
Serum LDH (IU/L)	390 (289.5–504)
Serum CRP (mg/dL)	6.823 (2.83–11.7)
Missing or unmeasurable	1
Serum KL-6 (U/mL)	556 (391–1032)
Missing or unmeasurable	4
Serum IgG (mg/dL)	948 (797–1145)
Missing or unmeasurable	10
Serum β-D-glucan (pg/mL)	34.8 (22.9–123.1)
Missing or unmeasurable	3
Treatments	
MTX use, number (%)	32 (86.5)
MTX dose (mg/week)	7 (4–9.5)
Missing or unmeasurable	1
PDN use, number (%)	25 (67.6)
PDN dose (mg/day)	4 (0–9)
Duration of PDN use (weeks)	16 (0–287)
Missing or unmeasurable	15
TAC/SASP/BUC/IGU use, number	4/2/6/6
IFX/ETN/GLM/TCZ/ABT use, number	2/2/1/4/3
TOF use, number	2
Comorbidities	
Diabetes mellitus, number (%)	8 (21.6)
Intistitial lung disease, number (%)	11 (29.7)
Emphysema, number (%)	4 (10.8)
Bronchiectasis, number (%)	3 (8.11)
LTBI, number (%)	2 (5.41)
Lung NTM, number (%)	1 (2.70)
Chemical prophylaxis	
SMX/TMP mixture, number (%)	0 (0)
Atovaquone, number (%)	3 (8.11)
Pentamidine, number (%)	0 (0)

Values indicate the median (interquartile range) unless otherwise mentioned.

**Table 2 ijms-24-07399-t002:** Treatments for PCP.

TMP/SMX mixture use, number (%)	35 (94.6)
Baseline TMP/SMX mixture dose (g/day)	9 (9–12)
Missing or unmeasurable	2
Pentamidine use, number (%)	12 (32.4)
Atovaquone use, number (%)	5 (13.5)
MPDN pulse therapy use, number (%)	12 (32.4)
PDN use, number (%)	35 (94.3)
Missing or unmeasurable	2
Baseline PDN dose (mg/day)	64 (40–80)
Missing or unmeasurable	2

Values indicate the median (interquartile range) unless otherwise mentioned.

**Table 3 ijms-24-07399-t003:** Clinical characteristics at baseline between PCP death patients and survival patients.

Characteristics	Death (*n* = 6)	Survivors (*n* = 31)	*p* Value
Age (years)	65.5 (59–73.5)	69 (63–75)	0.375
Female, number (%)	3 (50)	24 (77.42)	0.166
Disease duration (years)	7.67 (0.92–15.4)	10.8 (2.83–21.3)	0.314
Missing or unmeasurable	1		
Stage 1/2/3/4, number	(2/3/0/1)	(7/5/2/15)	0.236
Class I/II/III/IV, number	(0/5/1/0)	(7/17/4/1)	0.539
DAS28CRP	3.56 (3.55–4.29)	3.83 (2.73–4.62)	0.955
Missing or unmeasurable	3	15	
CDAI	8.6 (5.7–12.5)	8.55 (1.93–13.1)	0.85
Missing or unmeasurable	3	17	
SDAI	23.5 (5.70–24.5)	14.4 (8.13–19.8)	0.488
Missing or unmeasurable	3	17	
HAQ	1.88 (0.38–2.15)	0.75 (0–1)	0.237
Missing or unmeasurable	3	20	
Laboratory findings			
WBC (/μL)	9400 (4738–11,400)	7840 (5960–9800)	0.636
Lymphocytes (/μL)	950 (170–1716)	800 (580–1400)	0.757
Serum Alubmin (mg/dL)	2.75 (2.53–3.45)	3.3 (3–3.5)	0.238
Serum creatinine (mg/dL)	0.79 (0.66–2.63)	0.8 (0.59–0.97)	0.578
Serum LDH (IU/L)	455 (307–709)	360 (287–504)	0.293
Serum CRP (mg/dL)	15.9 (6.98–18.6)	5.7 (1.85–9.56)	0.035
Missing or unmeasurable	1		
Serum KL-6 (U/mL)	954 (529–1724)	499 (354–916)	0.118
Missing or unmeasurable		4	
Serum IgG (mg/dL)	874 (480–1249)	961 (797–1127)	0.671
Missing or unmeasurable	3	7	
Serum β-D-glucan (pg/mL)	77.8 (15.6–1626)	34.8 (23.2–77.0)	0.603
Missing or unmeasurable		3	
Treatments for RA			
MTX use, number (%)	4 (66.7)	28 (90.3)	0.177
MTX dose (mg/week)	5 (0–7)	8.0 (4.0–10.0)	0.086
PDN use, number (%)	6 (100)	19 (61.3)	0.064
PDN dose (mg/day)	9 (5.75–19.4)	3 (0–5)	0.006
Duration of PDN use (weeks)	158 (23–377)	12 (0–167)	0.222
Missing or unmeasurable	2	13	
TAC use, number (%)	0 (0)	4 (12.9)	0.352
SASP use, number (%)	1 (16.7)	1 (3.23)	0.183
bDMARDs/tsDMARDs use, number	1 (16.7)	13 (41.9)	0.243
Comorbidities			
Diabetes mellitus, number (%)	1 (16.7)	7 (22.6)	0.747
Interstitial lung disease, number (%)	1 (16.7)	10 (32.3)	0.444
Emphysema, number (%)	0 (0)	4 (12.9)	0.352
Bronchiectasis, number (%)	0 (0)	3 (9.68)	0.427
LTBI, number (%)	0 (0)	2 (6.45)	0.522
Lung NTM infection, number (%)	0 (0)	1 (3.23)	0.656

Values indicate the median (interquartile range) unless otherwise mentioned.

**Table 4 ijms-24-07399-t004:** Prognostic factors of death from PCP in patients with RA.

	Univariable Analysis			Multivariable Analysis		
Risk Factors	Crude HR	95% CI	*p*	Adjusted HR	95% CI	*p*
Age	0.978	0.908–1.076	0.611	0.93	0.839–1.050	0.213
Baseline serum CRP levels	1.094	0.990–1.188	0.075	1.103	0.968–1.249	0.127
Baseline PDN dose	1.193	1.066–1.340	0.003	1.23	1.448–0.813	0.003

The hazard ratios of death due to PCP were derived from univariable and multivariable Cox regression analysis. HR, hazard ratio; CI, confidence interval.

**Table 5 ijms-24-07399-t005:** Changes in treatment modalities for RA and chemical prophylaxis for PCP during the course after PCP treatment.

	Baseline (*n* = 37)	6 Months after (*n* = 24)	12 Months after (*n* = 21)	*p* Value (Initial vs. 6 months)	*p* Value (Initial vs. 12 Months)	*p* Value (6 Months vs. 12 Months)
Treatment modalities for RA						
MTX use, number (%)	32 (86.5)	12 (50)	10 (47.6)	0.002	0.002	0.317
MTX dose (mg/week)	7 (4–9.5)	1 (0–7.5)	1 (0–7)	<0.001	<0.001	0.471
PDN use, number (%)	25 (67.6)	20 (83.3)	15 (71.4)	0.011	0.096	0.157
PDN dose (mg/day)	4 (0–9)	5 (2–7.38)	5 (0–7)	0.856	0.814	0.035
bDMARDs/tsDMARDs use, number (%)	14 (37.8)	5 (20.8)	7 (33.3)	0.014	0.317	0.157
Modalities of bDMARDs/tsDMARDs	IFX 2, ETN 2, TCZ 4, GLM 1, ABT 3, TOF 2	ETN 1, ABT 4	ETN 2, ABT 4, TCZ 1	-	-	-
Chemical prophylaxis for PCP						
TMP/SMX mixture use, number (%)	0 (0)	14 (58.3)	12 (57.1)	<0.001	<0.001	1
TMP/SMX mixture dose (mg/week)	0 (0)	2.5 (0–7)	3 (0–7)	<0.001	<0.001	0.712
Atovaquone use, number (%)	3 (8.1)	3 (12.5)	1 (4.76)	1	0.317	0.317

Values indicate the median (interquartile range) unless otherwise mentioned.

**Table 6 ijms-24-07399-t006:** Changes in disease activities of RA during the course after PCP treatment.

	Baseline (*n* = 37)	6 Months after	12 Months after	*p* Value (Initial vs. 6 Months)	*p* Value (Initial vs. 12 Months)	*p* Value (6 Months vs. 12 Months)
DAS28CRP	3.58 (2.92–4.57)	2.48 (1.85–3.90)	1.91 (1.75–2.47)	0.046	0.001	0.271
Missing or unmeasurable	18	21	23			
CDAI	8.6 (2–12.15)	7.4 (1–9)	3 (2.1–8.8)	0.359	0.074	0.182
Missing or unmeasurable	20	22	24			
SDAI	16.05 (7.44–22.45)	7.4 (1.85–12.5)	3.3 (2.62–8.88)	0.188	0.012	0.285
Missing or unmeasurable	20	22	24			
CRP	6.83 (2.83–11.66)	0.26 (0.12–2.58)	0.18 (0.05–1.71)	<0.001	<0.001	0.098
Missing or unmeasurable	1	13	18			

Values indicate the median (inter quartile range) unless otherwise mentioned.

## Data Availability

Regarding the submission of raw data, because it is difficult to protect personal information, information will be disclosed only for reasonable offers. If any individual wants to request the data from this study, the contact should be the corresponding author.

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
