# Peer review of "Prognostic Factors Affecting Death in Patients with Rheumatoid Arthritis Complicated by Pneumocystis jirovecii Pneumonia and One-Year Clinical Course: The ANSWER Cohort Study"

_ijms, 2023, doi:10.3390/ijms24087399_

Round 1
Reviewer 1 Report
The paper by Shiba describes risk factors of fatal PCP in an adult RA cohort. The paper is well written and the conclusions are largely supported by the data. The paper could be improved several aspects.
1. The statement that “Pneumocystis jirovecii (PC) is a fungal pathogen that causes pneumonia in compromised hosts with decreased cell-mediated immunity” is not entirely true. The PERCH cohort study in Lancet in 2019 showed that Pneumocystis is a common cause of pneumonia in HIV – negative children without evidence of immunocompromised. It would be more accurate to state that Pneumocystis jirovecii (PC) is a fungal pathogen that causes pneumonia in infancy as well as in compromised hosts with decreased cell-mediated immunity.
2. There are several sentences in the abstract that have unclear meaning. “The usage rate and dose of methotrexate were significantly lower at 6 months and 12 months than at baseline. The usage rate of PDN significantly increased at 6 months compared with that at baseline, and its dose significantly decreased at 12 months after treatment compared with that at 6 months.” What are the significance of these data and why are they not emphasized more in the main text? It is unclear what the authors mean by baseline PDN dose. Is this the dose at PCP clinical presentation or the initiation dose for RA?
3. DAS28-CRP, CDAI, and SDAI and HAQ all need to be defined for a general readership audience.
4. Line 84, the authors refer to a diose of TMP/SMX of 9-12 mg/day. Which component TMP?
5. What was the effect of Atovaquone if any?
6. Other important clinical parameters that can affect PCP mortality need to be assessed including: duration of respiratory symptoms, Pa02/Fi02 ratio on admission (or pulse ox or 02 requirement), an serum LDH.
7. Which PCR assay was used? mtLSU?

Author Response
Responses to reviewer #1
Comment 1. The statement that “Pneumocystis jirovecii (PC) is a fungal pathogen that causes pneumonia in compromised hosts with decreased cell-mediated immunity” is not entirely true. The PERCH cohort study in Lancet in 2019 showed that Pneumocystis is a common cause of pneumonia in HIV – negative children without evidence of immunocompromised. It would be more accurate to state that Pneumocystis jirovecii (PC) is a fungal pathogen that causes pneumonia in infancy as well as in compromised hosts with decreased cell-mediated immunity.
Response: Thank you for helpful comment of reviewer#1, we collected sentence (page 2, line 45-46, ref 6).
Comment 2. There are several sentences in the abstract that have unclear meaning. “The usage rate and dose of methotrexate were significantly lower at 6 months and 12 months than at baseline. The usage rate of PDN significantly increased at 6 months compared with that at baseline, and its dose significantly decreased at 12 months after treatment compared with that at 6 months.” What are the significance of these data and why are they not emphasized more in the main text? It is unclear what the authors mean by baseline PDN dose. Is this the dose at PCP clinical presentation or the initiation dose for RA?
Response: As described in the discussion section (page 10, line 187- ), the use of MTX might have been avoided after PCP complication due to the risk of drug-induced pneumonia and to reduce the immunosuppressive state. In addition, it is considered that the use of PDN as an anti-inflammatory therapy for PCP increased after PCP complication. As reviewer #1 pointed out, changes in the use of MTX and PDN after PCP complication can be reasonably predicted in clinical practice, and have been removed from the abstract due to their low importance. In addition, in order to adjust the content and volume of the abstract, we added the results of univariate analysis that comparison of the baseline clinical characteristics of patients who died from PCP and survivors (page 1, line 29-30).
The present study suggested that the use of high-dose corticosteroids for RA treatment leads to poor prognosis when PCP complicated, and the abstract text has been corrected (page 1, line 33-34).
Comment 3. DAS28-CRP, CDAI, and SDAI and HAQ all need to be defined for a general readership audience.
Response: Thank you for helpful comment of reviewer#1, DAS28-CRP, CDAI, SDAI and HAQ have been added with more detailed explanations for a general readership audience (page 12, line 280-290).
Comment 4. Line 84, the authors refer to a diose of TMP/SMX of 9-12 mg/day. Which component TMP?
Response: Thank you for helpful comment of reviewer#1, we used TMP/SMX mixture and 1 mg of TMP/SMX contains 80 mg of TMP and 400 mg of SMX, so we corrected and added (page 2, line 76. page 3, line 80, 81, Table 1, 2, 5).
Comment 5. What was the effect of Atovaquone if any?
Response; As Reviewer#1 pointed out, prophylactic administration of atovaquone is useful for preventing PCP complications, but it is not possible to completely eliminate PCP complications, so the following sentence has been added to the discussion (page 9, line 170-174, ref 20): Of 107 Japanese CTD patients receiving high-dose corticosteroids who were prophylactically treated with atovaquone, 5 were reported to have complicated PCP [20]. Although the frequency may be low, it is also considered necessary to pay attention to the possibility of complication with PCP even during prophylactic administration of atovaquone in patients with RA under immunosuppressive therapy.
Comment 6. Other important clinical parameters that can affect PCP mortality need to be assessed including: duration of respiratory symptoms, Pa02/Fi02 ratio on admission (or pulse ox or 02 requirement), an serum LDH.
Response: Following the comment of reviewer#1, we extracted the respiratory conditions including duration of respiratory symptom and PaO2/FiO2 ratio at baseline, and compared these between the PCP death group and the survival group. As a result, the PaO2/FiO2 ratio was significantly lower in the PCP death group compared with the survival group. We added in text (page 11, line 267-269. page 6, line 101-108).
Comment 7. Which PCR assay was used? mtLSU?
Response: Following the comment of reviewer#1, we added PCR assay method (page 11, line 259-260).

Reviewer 2 Report
I read this manuscript with interest and this should add value to the literature. However, several major comments should be addressed.
Introduction:
1. The last sentence in the introduction is more a method or objective, therefore, a clear aim statement should be written.
2. Please, see the last comment about the references.
Results:
1.
2.1. Characteristics of the Patients: The text in this this section is encouraged to be less, because it replicate what is available in the text.
2. Write the abbreviations in the Table 1 legend, they are mentioned in the methods at the end of the manuscript.
3. In the legend of table 1 "Values indicate the median (interquartile range)" it is not precise: write for example ....unless otherwise mentioned or put a star or a mark etc.
4. In Table 1 and the text before it, what you mean with Complications? I supposed the more accurate term is diseases or comorbidities etc, as you descrpe the baseline data.
5. Table 3 legend: ". *P <0.05, **P <0.01." there is no need for this basic data as you put the original value (the same applied for table 4).
6. The statistical insignificance was expected in Table 3 as the survivors group is 5 times more than the non-survivors. This raise a point of finding another outcome other than the death.
7. What about the duration of prednisone treatment? this would be more important than high dose for short term use...
8. In section "2.6. Changes in Disease Activities of RA During The Course After PCP Treatment ", 4 decimal points of p values appear, please, be consistent during the whole manuscript.
9. The results section does not show data about the missing data, which is, of course, expected as this study performed retrospectively.
10. Despite being multicenter and over several years, the sample size is small
Methods:
L234-236: unclear sentence about the informed consent
Discussion:
The discussion should be enhanced. The findings should be compared with new studies (see below comment). More focus on the corticosteroid use is highly encouraged by comparing with other studies, not only limited to RA. This risk factor is used in other diseases like asthma and COPD exacerbations, COVID-19, cancer, etc.
A high percentage of the cited references are very old; more than 10 years ago (20/29 references). I strongly recommend updating the references and make extensive literature review. I suggest the following (not limited to) very recent reviews/articles:
Alsayed AR, Al-Dulaimi A, Alkhatib M, Al Maqbali M, Al-Najjar MA, Al-Rshaidat MM. A comprehensive clinical guide for Pneumocystis jirovecii pneumonia: a missing therapeutic target in HIV-uninfected patients. Expert Review of Respiratory Medicine. 2022 Dec 14:1-24.
Lang Q, Li L, Zhang Y, He X, Liu Y, Liu Z, Yan H. Development and Validation of a Diagnostic Nomogram for Pneumocystis jirovecii Pneumonia in Non-HIV-Infected Pneumonia Patients Undergoing Oral Glucocorticoid Treatment. Infection and Drug Resistance. 2023 Dec 31:755-67.
Alsayed, A.R., Talib, W., Al-Dulaimi, A., Daoud, S. and Al Maqbali, M., 2022. The first detection of Pneumocystis jirovecii in asthmatic patients post-COVID-19 in Jordan. Biomolecules and Biomedicine, 22(5), pp.784-790.
Author Response
Responses to reviewer #2
Comment 1. The last sentence in the introduction is more a method or objective, therefore, a clear aim statement should be written.
Response: Thank you for helpful comment of reviewer#2, at the end of the introduction section, we added a sentence that clearly states the aim of the present study (page 2, line 60-62).
Comment 2. Characteristics of the patients: The text in this this section is encouraged to be less, because it replicate what is available in the text.
Response: Following the helpful comment of reviewer#2, since the details are listed in table 1, we have summarized them in short sentences.
Comment 3. Write the abbreviations in the Table 1 legend, they are mentioned in the methods at the end of the manuscript.
Response: Following the helpful comment of reviewer#2, we have summarized the abbreviations at the end of the manuscript.
Comment 4. In the legend of table 1 "Values indicate the median (interquartile range)" it is not precise: write for example ....unless otherwise mentioned or put a star or a mark etc.
Response: Following the helpful comment of reviewer#2, we changed the sentence in the legend of tables.
Comment 5. In Table 1 and the text before it, what you mean with Complications? I supposed the more accurate term is diseases or comorbidities etc, as you descrpe the baseline data.
Response: Following the helpful comment of reviewer#2, we changed the word from “complications” to “comorbidities” in the text (page 2, line 71), table 1, and table 3.
Comment 6. Table 3 legend: ". *P <0.05, **P <0.01." there is no need for this basic data as you put the original value (the same applied for table 4).
Response: Following the helpful comment of reviewer#2, the p-values ​​in the tables (table 3, table 4, table 5, table 6) are written as original values, and legends are omitted.
Comment 7. The statistical insignificance was expected in Table 3 as the survivors group is 5 times more than the non-survivors. This raise a point of finding another outcome other than the death.
Response: As reviewer#2 pointed out, the survivor group was skewed at five times the number of non-survivors, which could confound the results. It was added to the limitation as a future issue that requires further consideration with an increased number of cases (page 10, line 222-224).
Comment 8. What about the duration of prednisone treatment? this would be more important than high dose for short term use...
Response: Following the comment of reviewer#2, we extracted the duration of prednisolone treatment at baseline, and compared these between the PCP death group and the survival group. As a result, there was no difference between the two groups. We added them in table 1 and table 3.
Comment 9: In section "2.6. Changes in Disease Activities of RA During The Course After PCP Treatment ", 4 decimal points of p values appear, please, be consistent during the whole manuscript.
Response: Following the helpful comment of reviewer#2, we standardized p-values ​​to 3 decimal points throughout whole manuscript and tables.
Comment 10; The results section does not show data about the missing data, which is, of course, expected as this study performed retrospectively.
Response: As reviewer#2 pointed out, this study is a multicenter study and there are missing or unmeasurable data due to limitations in data collection. Following the helpful comment of reviewer#2, we added missing or unmeasurable data in tables.
Comment 11. Despite being multicenter and over several years, the sample size is small
Response: As Reviewer #2 pointed out, the sample size of the present study is small, and it is possible that the spread of chemical prophylaxis against PCP has reduced the number of cases that develop PCP compared to the past. So, we added it to the limitation as a future issue that requires further consideration with an increased number of cases (page 10, line 218, 223-224).
Comment 12. L234-236: unclear sentence about the informed consent.
Response: As Reviewer #2 pointed out, we corrected sentence more clearly about the informed consent as following (page 11, line 247-248): At institutes other than Osaka University Hospital, written informed consent was obtained from the participants. The Osaka University Hospital Ethics Committee board waived the requirement for patient informed consent because of the anonymous nature of the data.
Comment 13:The discussion should be enhanced. The findings should be compared with new studies (see below comment). More focus on the corticosteroid use is highly encouraged by comparing with other studies, not only limited to RA. This risk factor is used in other diseases like asthma and COPD exacerbations, COVID-19, cancer, etc.
Response: Following the helpful comment of reviewer#2, corticosteroid use has also been reported as a risk factor for PCP complications in other diseases, so it is suggested that reducing the amount and duration of corticosteroid use may reduce the complication rate of PCP and improve the prognosis. We added this in discussion section (page 10, line 182-186, ref 21-26).
Comment 14. A high percentage of the cited references are very old; more than 10 years ago (20/29 references). I strongly recommend updating the references and make extensive literature review. I suggest the following (not limited to) very recent reviews/articles:
Response: Following the helpful comment of reviewer#2, we updated what could make into new references including recommendation by reviewer#2 without changing the content of the manuscript.

Round 2
Reviewer 2 Report
The authors addressed the comments and the manuscript is now better